# Case Fatality as an Indicator for the Human Toxicity of Pesticides—A Systematic Scoping Review on the Availability and Variability of Severity Indicators of Pesticide Poisoning

**DOI:** 10.3390/ijerph18168307

**Published:** 2021-08-05

**Authors:** Susanne Moebus, Wolfgang Boedeker

**Affiliations:** 1Institute for Urban Public Health, University Hospital, University of Duisburg-Essen, 45128 Essen, Germany; Susanne.Moebus@uk-essen.de; 2EPICURUS—Impact Assessment, 45136 Essen, Germany

**Keywords:** pesticides, case-fatality, poisoning, highly hazardous, severity scores

## Abstract

Objective: To investigate if case fatality and other indicators of the severity of human pesticide poisonings can be used to prioritize pesticides of public health concern. To study the heterogeneity of data across countries, cause of poisonings, and treatment facilities. Methods: We searched literature databases as well as the internet for studies on case-fatality and severity scores of pesticide poisoning. Studies published between 1990 and 2014 providing information on active ingredients in pesticides or chemical groups of active ingredients were included. The variability of case-fatality-ratios was analyzed by computing the coefficient of variation as the ratio of the standard deviation to the mean. Findings: A total of 149 papers were identified of which 67 could be included after assessment. Case-fatality-ratio (CFR) on 66 active ingredients and additionally on 13 groups of active ingredients were reported from 20 countries. The overall median CFR for group of pesticides was 9%, for single pesticides 8%. Of those 12 active ingredients with a CFR above 20% more than half are WHO-classified as “moderately hazardous” or “unlikely to present acute hazard”. Two of seven pesticides considered “unlikely to present hazard in normal use” showed a CFR above 20%. The cross-study variability of reported case fatality was rather low. Studies most often utilized the Glasgow Coma Score for grading the severity of poisoning. Conclusion: Although human pesticide poisoning is a serious public health problem, an unexpectedly small number of publications report on the clinical outcomes within our study period. However, CFRs of acute human pesticide poisoning are available for several groups of pesticides as well as for active ingredients showing moderate cross-study variability. Our results underline that CFR is an indicator of the human toxicity of pesticides and can be utilized to prioritize highly hazardous pesticides especially since there is limited correspondence between the animal-test-based hazard classification and the human CFR of the respective pesticide. The reporting of available poisoning data should be improved, human case-fatality data are a reasonable tool to be included systematically in the periodic statutory review of pesticides and their regulation.

## 1. Introduction

Pesticides have become a major input in the world’s agriculture over the last decades. Usually addressed as insecticides, herbicides, rodenticides, or fungicides according to their overall target species they are mainly deployed to efficiently increase crop production. Furthermore, they are considered beneficial from a public health perspective because by definition they are also used to control vectors of human diseases. However, pesticides comprise a great variety of chemical groups with partly general mechanisms of action. Their detrimental effects are often unspecific and affect non-target organisms including humans.

As early as 1990, a task force of the World Health Organization (WHO) estimated that about one million unintentional pesticides poisonings occur annually leading to approximately 20,000 deaths [1]. Additionally, two million cases were expected to follow from self-harm. It was recognized that low-income countries and countries in transition were particularly affected by the impact of pesticide poisoning and actual numbers were probably much higher as many cases remain unreported.

It took more than 30 years for an updated estimation to be published, finding that worldwide annually 385 million cases of unintentional acute pesticide poisoning are expected to occur, including around 11,000 fatalities [2]. Additionally, 110,000–168,000 fatalities were lately estimated to occur from suicidal pesticide poisoning worldwide mostly in rural agricultural areas in low- and middle-income countries [3]. During the last decades, international bodies have taken up the issue of health hazards from pesticide exposure and adopted a great number of resolutions and programs to improve their safe use [4]. Realizing, however, that despite all efforts there might be no safe use of toxic pesticides especially under conditions of poverty, the International Code of Conduct on Pesticide Management of WHO and the Food and Agriculture Organization (FAO) endorses a new policy approach by considering the prohibition of highly hazardous pesticides [5].

Pesticides are considered highly hazardous when presenting high acute toxicity according to internationally accepted classification systems such as the WHO Recommended Classification of Pesticides by Hazard [6]. In addition, pesticides that cause severe or irreversible harm to health “… under conditions of use in a country” may be considered as highly hazardous [5].

The WHO Recommended Classification of Pesticides by Hazard [6] is in widespread use for hazard identification and risk management. This classification is based primarily on estimates of the acute oral and dermal toxicity to rats. Based on its acute toxicity on rats a pesticide is assigned to five toxicity classes. It is assumed that these hazard classes also capture the acute toxicity for humans. However, an assignment to a higher or lower class is possible if the active ingredient is proved to be more or less toxic in humans [6]. Unfortunately, studies have reported poor agreement between the acute human toxicity of pesticides and the respective WHO hazard classes [7,8,9].

Case fatality and severity scores might be a more realistic indicator for human toxicity of substances than hazard classes based on animal testing. If so, the indicators should primarily point to the substance-specific toxic properties and not on the characteristics of the incident and treatment e.g., cause, dose, and time lag between exposition and treatment. A low variability of the e.g., pesticide specific case-fatality-rate (CFR) would then indicate problematic chemicals from a public health perspective as the human toxicity of an agent in general was captured rather than the clinical course of a specific poisoning. Recently, it has been suggested that a case fatality after self-poisoning greater than 5% should be used as an indicator of a highly hazardous pesticide and that a complete ban of these pesticides be targeted [10].

In emergency medicine, several scores and classification systems have been introduced to predict the fatality of a disease and allow for risk stratification [11]. The International Program on Chemical Safety in cooperation with the European Community and the European Association of Poisons Centers and Clinical Toxicologists have introduced and encouraged the use of the Poison Severity Score (PSS) for the prognostic assessment of poisonings and the selection of treatment [12]. Case-fatality is used as an end-stage category in the PSS and other severity classifications.

In order to study whether and which indicators of the severity of poisonings can be used to prioritize pesticides of public health concern we systematically reviewed the scientific literature. We aimed at answering the following research questions:For which active ingredients in pesticides or for which group of pesticides have human case- fatality-ratios been published?What is the geographical distribution and the variability of the reported case-fatality ratios?What is the relationship between the human case-fatality and WHO hazard classes?Which factors influence the case-fatality?Which severity scores are used with respect to pesticide poisonings?

Given these research questions our study was rather scoping. Using, e.g., case-fatality as toxicity indicator in hazard assessment presumes that data were available for a sufficient number of pesticides and that severity was validly estimated from accessible sources. We therefore described heterogeneity across countries, cause of poisonings, and treatment facilities.

## 2. Methods

We conducted a systematic literature review without prior protocol by starting the search for publications in the database PUBMED. We used the terms “pesticides” AND (“case-fatality-ratio” OR “case-fatality-rate” OR “poisoning severity score”) and allowed for studies in English, German, Portuguese, and Spanish with a publication date between January 1990 and October 2014. The search procedure was repeated with the database SCOPUS which has a higher coverage outside medical sciences and includes the database EMBASE completely as of 1996. In a sensitivity analysis addressing our search strategy and a possibly too strict selection, we additionally searched for specific pesticides and checked these with the results in our automatic search as outlined below.

According to the PRISMA-Statement (prisma-statement.org) all records were screened and excluded in case abstracts clearly indicated non-eligibility, e.g., when only specific symptoms of poisonings or animals were studied. Full-text-analysis was carried out on all other records. Studies were considered eligible when addressing active ingredients in pesticides or groups of active ingredients (e.g., organophosphates). Case-studies and papers which do not report or not allow to calculate a case fatality were excluded. The search was supplemented by inspecting bibliographic reference lists in all identified papers. Articles initially not identified by the automatic search were then manually back searched. Finally, 149 papers were identified of which 67 could be included after assessment. We excluded 25 papers by abstract and 57 by full-text analysis mainly because no information on active ingredients or group of pesticides was presented (Figure 1).

Case fatality was extracted from all included publications. In most papers, case fatality was referred to as case-fatality-ratio (CFR) or mortality-ratio and was given as number of fatal poisonings divided by the number of all poisonings with a specific agent or group of agents respectively. When the CFR was not stated in the studies that we calculated from given numbers of incidents. Case fatality and indicators used as descriptors of the poisoning severity as well as information on the number of patients, the country, year, cause of poisoning, and timespan of the study were retrieved study-wise for each poisoning agent in a data base.

Overall, case numbers and CFR were studied by minimum, mean, median, and maximum values. The cross-study variability of the CFR was assessed by the coefficient of variation (CV) as the ratio of the standard deviation to the mean and its normalized form which limits the CV to the interval 0–1 and adjusts for the number of observations [13]. Mathematically, a CV lower or equal 100% indicates low variability taking the exponential distribution as a reference. Calculation of CV was restricted to those pesticides which were addressed in more than three papers. All calculations were done with SAS statistical software, Version 9.4 (SAS Institute Inc., Cary, NC, USA).

Due to our scoping interest, we did not impose any quality constraints on the studies. We furthermore did not weight the extracted data with respect to study characteristics. The search strategy was developed by both authors. The screening of records and data retrieval was mainly done by one author (WB). To assess possible bias from handling of exclusion criteria a random sample of 20% of all excluded records were additionally cross-checked by the other author (SM). SM furthermore repeated the data extraction of a 20% sample of all included papers. There was no disagreement between both authors’ decisions.

## 3. Results

Sixty-seven publications [7,8,9,14,15,16,17,18,19,20,21,22,23,24,25,26,27,28,29,30,31,32,33,34,35,36,37,38,39,40,41,42,43,44,45,46,47,48,49,50,51,52,53,54,55,56,57,58,59,60,61,62,63,64,65,66,67,68,69,70,71,72,73,74,75,76,77] reporting case fatality rates on 66 active ingredients and additionally on 13 groups of active ingredients were identified (Table 1). Moreover, 58% of the active ingredients are covered by just one publication. The most mentioned active ingredient is glyphosate which is addressed in 16 papers followed by paraquat in 14 papers. With respect to groups of pesticides organophosphates are most frequently studied. Thirty-one papers report on studies from 14 different countries. In total, 20 countries are covered by the included studies with Taiwan and Sri Lanka most often addressed (Table 2).

The overall median CFR for the listed groups of pesticides is 9.2% with no reported fatalities for coumarins and pyrethrins (Table 1). The highest group related median CFR is 38% for diethyl-organophosphates. With respect to active ingredients the median CFR is 8.2% with highest CFR of 100% seen for propamocarb followed by parathion methyl with 60%.

Of the active ingredients considered, 17% show case-fatality above 20. Table 3 shows how the CFR is captured by the WHO Recommended Classification of Pesticides by Hazard.

Of those 11 active ingredients with a CFR above 20% two (parathion ethyl and parathion methyl) are WHO-classified as extremely hazardous (Ia) and two (dichlorvos, monocrotophos) as highly hazardous (Ib). A CFR above 10% was observed in 38% of pesticides which are WHO-classified as Ia or Ib. Furthermore, two of seven pesticides considered “unlikely to present hazard in normal use” show CFR above 20% (picloram, propamocarb). However, propamocarb fatality was reported in one study only with a single patient [8]. Figure 2 displays CFR for active ingredients and their hazard classes.

Thirty-two pesticide groups or active ingredients were studied by more than one paper and nine by more than three papers. Table 4 gives the coefficients of variation (CV) along with case-fatalities and WHO-classification on all pesticides addressed in more than three publications. All CFR show a coefficient of variation lower or equal 100%. Four out of the seven active ingredients are seen with a CV even well below 100% whereas only malathion and glyphosate reach 93% and 100%, respectively. Glyphosate serves as an example of how study characteristics impact on the variability of CFR. The highest value of 29% was seen in a study in Taiwan [38] recruiting in two hospitals which serve as referral hospitals and include a poisoning control center. The lowest value of 0.06% follows from two deaths in 3464 human exposure cases (98% unintentional) collected in the US National Poison Data System by telephone calls received in 57 regional poison centers [48]. If only those cases treated in health care facilities were taken as the denominator (see Mowry et al. p 1165), the CFR would be 3.6% and further decrease the variability across countries. With respect to the normalized coefficient of variation, which adjusts for the number of publications, malathion is the active ingredient with the highest variability but also well below the possible 100%.

The variability of CFR might depend on the cause of poisoning. Most pesticide poisonings reported in the included papers follow from suicidal intention. Therefore, a direct comparison of accidental and suicidal causes is possible only for organophosphates with two papers on accidental poisoning [42,44], five on suicidal poisoning [8,18,31,33,53], and two on both [30,75]. When intoxication was suicidal, mean CFR was 14% compared to 6% for unintentional occasions.

Clinical indicators for the severity of poisonings were mentioned in many papers. Although the Poisoning Severity Score (PSS) was part of the search terms we found more papers reporting on the Glasgow Coma Score (GCS). Additionally, the Acute Physiology and Chronic Health Evaluation Score (APACHE), the Sequential Organ Failure Assessment (SOFA), and the Simplified Acute Physiology Score (SAPS) were used along with scores built specifically by the study authors (Table 5). Studies often aim at a comparison of different indicators with respect to their performance for predicting study specific clinical outcomes. More information on the used indicators is available in some studies, e.g., mean and grading of scores. However, the number of papers in this review is too limited to study the variability of indicators with respect to specific group of pesticides or active ingredients.

## 4. Discussion

Case fatality of poisoning is considered as basically dependent on intention, dose, time between exposition and treatment, and access to treatment. Therefore, the CFR is not expected to indicate toxicity of an agent but of the specific poisoning case. In contrast, our results show that variability across countries and studies is rather low even when group of pesticides with different chemical compounds are considered. This agrees with Hrabetz et al. [33], who concluded that the CFR in a German cohort of organophosphate intoxicated patients were similar compared to respective rates in developing countries. In contrast, Eddleston et al. [28] suggest that CFR for self-poisoning—including pesticides—in rural Sri Lanka may be ten-fold higher than that of England. However, this comparison was not pesticide specific and therefore does not account for different substance usage.

The cause of poisoning seems to influence the CFR. We calculated a mean CFR for organophosphate poisoning more than twice as high for suicidal intoxication compared to unintentional incidents. This relation has also been analyzed in other studies. Recena and colleagues [54] report for the region of Mato Grosso, Brazil, a CFR of 27.5% in suicidal poisoning compared to 13% for all pesticide poisonings. An even higher relation of 9.1% to 3% was observed Brazil-wide. With respect to single active ingredients a high relation was found especially for some insecticides. Chen et al. [21] calculated an odds ratio of 2.3 for severe poisoning by suicidal versus unintentional cause controlling for several factors in multiple regression. Taken together, the observed low trans-country variability of CFR seen in this review might reflect that most of the reported pesticides poisonings result from intentional ingestion.

With respect to the WHO Recommended Classification of Pesticides by Hazard, our review shows less agreement between WHO classes and CFR than expected. Only about one third of the pesticides with highest CFR (>20%) are marked by the highest WHO class (Ia, Ib) and about one fifth are even considered as unlikely to present acute hazard. This disagreement confirms what previously has been reported by single studies [7,8,9]. Given our data and based on the median CFR, 40 of 66 active ingredients exhibit a CFR above 5%, which has been proposed a cut-off to indicate highly hazardous pesticides [10].

Our scoping review has some limitations. Although single case studies were excluded from our review, some CFRs are based on a small number of cases and should be handled with caution. The number of pesticides addressed by several studies is too small in our review to allow for an in-depth analysis of variability of CFR with respect, e.g., to WHO hazard classes or the number of cases. Furthermore, we are not aware of publications comparing human CFRs of pesticide poisoning with toxicity outcomes of animal experiments. However, case-study based human lethal doses of substances have been studied for interspecies comparisons. Ekwall et al. [78] compiled lethal doses from medical handbooks for 50 selected chemicals including five pesticides. The mean lethal doses stretch from 2.5 g for paraquat to 52 g for malathion with little variation within the active ingredients. A low variability of CFR would correspond to a low variability of lethal doses.

The reliability of CRF may be limited by selection bias; say that primarily severe poisonings were subject to treatment and therefore a higher case fatality is observed in studies. We tried to avoid this possible bias by excluding pure case studies from this review. All CFRs presented are based on a longitudinal study design (retro- or prospective). In most studies, all patients with poisonings in a given time period admitted to the hospitals were included by study design so that CFR is indicative for all poisonings treated. However, it can still be the case that a hospital or center is specialized to the treatment of poisonings and therefore primarily severe cases will be admitted here. Eddleston et al. [28] analyzed the influence of patient transfers between hospitals and found that 50% of self-poisonings admitted to small rural hospitals in Sri Lanka were treated there and discharged home. So, CFR in secondary referral hospitals were high because of selection bias. However, their observed CFR of 7.4% in rural and 11% in secondary hospitals show no substantial difference.

There is also concern for the underestimation of CFR as it can be assumed that hospitals prepared and interested in conducting and publishing scientific studies probably have much higher standards in treatment facilities and staff. Dharmani and Jaga [79] reviewed the literature on organophosphate poisoning und summarized that in rural areas of developing countries access to immediate treatment is often limited. Furthermore, rural hospitals suffer from poor equipment (e.g., missing ventilation). A further reason for underestimation of CFR may be that patients die before admission to medical services and therefore stay unconsidered in CFR figures of hospitals statistics [28,62].

Time to admission does not seem to play a clear-cut role for pesticide poisonings. No significant difference between survivors and non-survivors with respect to time from ingestion was observed in several studies [7,39,47,62]. In contrast, Vucinic et al. [71] found late admission >4 h to hospital a risk factor for mortality of carbamate poisoning. Chen et al. [21] showed an increasing percentage of severe/fatal glyphosate poisoning with increasing time lag until admission even after adjustment in multivariate analysis. However, for glyphosate it was reported that non-survivors were hospitalized significantly faster than survivors [39], the same was also observed for paraquat [73]. Therefore, the influence of time to admission on CFR may be not only specific for active ingredients, but also arise from selection bias as patients may die before reaching hospitals [7].

Finally, the search strategy was selective as the search term “pesticides” was used. This is a trade-off between a high number of false positives references when merely “poisonings” were addressed and the high number of singles active ingredients in pesticides which could be included in the search. In order to rate a possibly too rigid selection, we conducted an additional search as a sensitivity analysis. We selected all pesticides of the PAN (Pesticide Action Network)-International list of highly hazardous pesticides [80] which are denoted by three or more items of concern. Ten out of these 15 active ingredients were so far not addressed by the studies included in this review (azinphos-methyl, captafol, chlordane, DDT, hexachlorobenzene, lambda-cyhalothrin, omethoate, parathion, PCP (pentachlorphenol), phosphamidon). Each of these pesticides was searched in combination with “case-fatality *” in the data bases Pubmed, Embase and Scopus with no language restraints. As a result, 16 studies could be identified which were not found by our original search strategy for the above mentioned period. However, none of these studies were eligible given our outlined criteria. Reasons were e.g., different scope of studies (e.g., Malaria and DDT) or patchy acronyms (e.g., PCP short for “pentachlorphenol” as well as “pneumocystis pneumonia”). Additionally, two papers were identified which were published after our inclusion date. Both were also not eligible for our review.

Summing up, the apparently good coverage of our search strategy is probably due to the rather extensive back search which identified more records than the automatic search. Still, we assume that a great number of eligible studies remain undetected because active ingredients of pesticides are not mentioned as keywords or in the title. Additionally, pesticides are not a well-defined group of chemicals and names often vary. However, we do not imagine reasons for a systematic bias in reported CFR due to studies which slipped our search. In fact, after finishing this review we have been made aware of additional studies giving CFR well within the range of Table 1 [81,82,83]. However, for aluminum phosphide a CFR of 77% was reported [84] higher than that in Table 1 (max = 67%). In a review on diquat poisoning, Jones and Vale reported 30 cases in the timespan 1968–1999 with a CFR of 43% [85]. This contrasts to our results of no fatalities (Table 1) reported for 56 cases treated in US health centers in 1992 [48]. A fatal poisoning by fipronil is stated by [86] as a historic case with no details on the inclusion criteria. The case fatalities ratios concerning a greater number of active ingredients varied from 0% to 42% (paraquat) in a study analyzing cohort data of pesticide self-poisoning from Sri Lanka [10]. However, these data have partly been published before [8] and are included in our review. Our review certainly omitted newly authorized pesticides because human case-fatality studies usually are carried out after longer-term use.

Only a small number of active ingredients in pesticides were addressed by several studies. As such, our analysis could not be extended to possible constituents of case-fatality like dose, populations, chemical formulations of the pesticides, or treatment regimes. Furthermore, we refrained from imposing quality scores to the study results prior analysis. This might be possible after a more comprehensive survey and documentation of poisoning is achieved. Identification of factors influencing the CFR can further be supported by analytical approaches [87]. This might assist to objectively link exposure data to symptoms and allows mitigating factors like protective clothing and handling procedures as well as quality and stability of pesticide formulations to be taken into account. However, if human case-fatality was to be used as an indicator of the toxicity of substances in hazard classification then non-stratified rates would be needed because rather general assessments were intended.

This study derived from a project on the assessment of pesticide poisoning in Germany [88,89]. An earlier version has been published on bioRxiv.org.

## 5. Conclusions

We studied the availability and variability of case-fatality ratios due to pesticide poisonings. Given the large number of world-wide pesticide poisonings we found an unexpectedly small number of 67 publications on 66 active ingredients covering 20 countries. Besides missed publications this is probably best explained by insufficient resources of primary care hospitals to systematically analyze and report poisoning cases. Furthermore, it is well known that poison control centers in many countries do not provide public reporting at all or report on an aggregated level only.

This review confirms the limited agreement between the case-fatality-ratio of human acute pesticides poisoning and the WHO hazard classification of the respective pesticide. The hazard and risk assessment of the acute toxicity of pesticides should not be based on data from animal tests only but should also consider the available information on human intoxications.

The active ingredient specific case-fatality-rate of acute human pesticide poisoning showed moderate variability in this systematic scoping review. The case-fatality therefore seems to indicate not only the severity of an individual poisoning incident but also an intrinsic property of the pesticide. So, the case-fatality might well capture the human acute toxicity of an active ingredient and could be utilized for prioritization of highly hazardous pesticides. By simply improving the reporting of available poisoning data valuable indicators could be gained for hazard and risk management of pesticides. As a policy implication, human case-fatality data should be retrieved along with the animal-test based hazard classes as part of the periodic statutory review of pesticides and their regulation.

## Figures and Tables

**Figure 1 ijerph-18-08307-f001:**
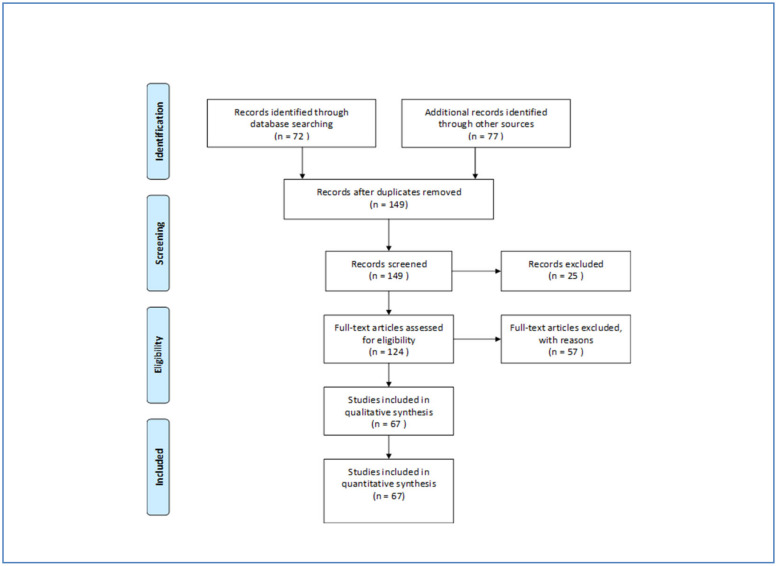
Flow chart of selection procedure and search results.

**Figure 2 ijerph-18-08307-f002:**
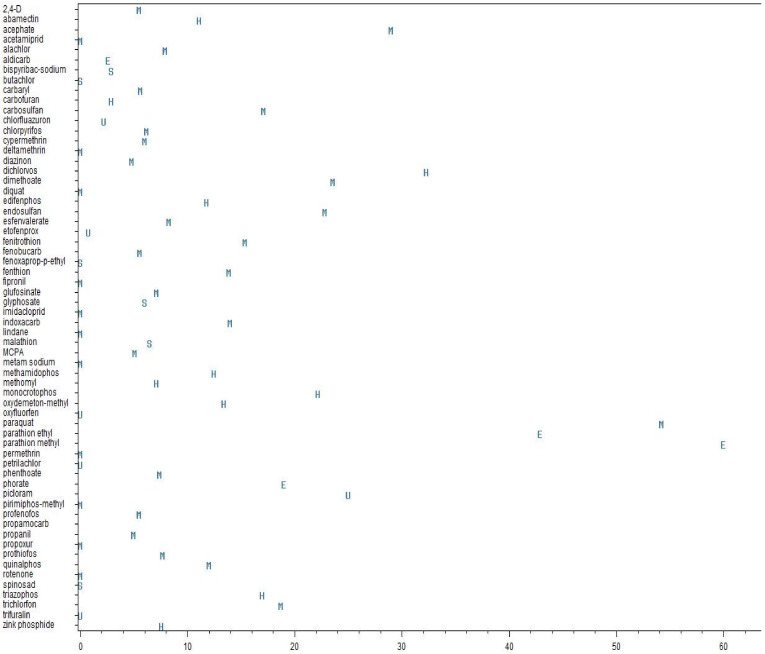
Median case-fatality-ratios (%) for active ingredients of pesticides and WHO hazard classes. E = extremely hazardous, H = highly hazardous, M = moderately hazardous, S = slightly hazardous, U = unlikely to present acute hazard. Some pesticides with high human case-fatality are inappropriately classified. Agents stated “obsolete” or “VF” not shown.

**Table 1 ijerph-18-08307-t001:** Study characteristics and case fatalities for reported groups of pesticides and active ingredients.

Group of Pesticide	Publications	Countries	Case-Fatality-Ratio (%)			Cases (*n*)			Severity Indicator ^1^	Country	Reference
	*N*	*n*	Median	Min	Max	Median	Min	Max			
carbamates	9	6	5.1	0.0	14.2	60	6	1433	CFR	Brazil, India, Israel, Serbia, Sri Lanka, Taiwan	[7,8,18,28,43,45,69,71,76]
carbamates/OP ^2^	1	1	5.0			280			PSS, CFR	Brazil	[17]
chloracetanilide	1	1	3.6			28			CFR, PSS	Korea	[58]
coumarin	1	1	0.0			82			PSS, CFR	Brazil	[17]
cyanide	1	1	24.1						CFR	Taiwan	[76]
diethyl-OP	1	1	38.0			8			CFR, PSS, APACHE, SOFA, GCS	Germany	[33]
dimethyl-OP	1	1	11.0			19			CFR, PSS, APACHE, SOFA, GCS	Germany	[33]
fungicides	1	1	6.1			49			CFR	Sri Lanka	[8]
herbicides	1	1	12.4			2783			CFR	Sri Lanka	[8]
organochlorines	2	2	18.4	16.7	20.0	112	12	212	CFR	India, Sri Lanka	[7,8]
organophosphate	31	14	11.1	2.9	73.0	94	16	5226	CFR, APACHE, PSS, SOFA, GCSCFR, SAPS, CFRCFR, SAPSII	Australia, China, Germany, India, Iran, Israel, Japan, Jordan, Slovenia, SouthAfrica, Sri Lanka, Taiwan, Turkey, Zimbabwe	[7,8,14,15,16,18,19,22,23,25,30,31,32,33,36,42,44,53,56,57,60,64,65,66,69,70,72,74,75,76,77]
pyrethrins	1	1	0.0			5522			CFR	USA	[48]
pyrethroids	3	3	0.7	0.0	1.0	203	140	23,853	PSS, CFR	Brazil, Sri Lanka, USA	[8,17,48]
**Active Ingredients**											
2,4-D	1	1	5.5			20			CFR	Brazil	[54]
abamectin	1	1	11.1			18			CFR	Sri Lanka	[8]
acephate	1	1	29.0			14			CFR	India	[7]
acetamiprid	1	1	0.0			11			CFR	Sri Lanka	[8]
alachlor	2	2	8.0	4.8	11.1	36	9	63	CFR	Sri Lanka, Taiwan	[8,46]
aldicarb	2	2	2.6	0.0	5.2	37	35	39	CFR, PSS	France, USA	[51,52]
aldrin	1	1	13.3			49			CFR	Brazil	[54]
aluminium phosphide	2	1	48.9	31.0	66.7	255	39	471	CFR, APACHE, SAPS, GCS	Iran	[61,62]
bispyribac-sodium	1	1	2.9			103			CFR	Sri Lanka	[8]
butachlor	1	1	0.0			70			CFR	Taiwan	[46]
carbaryl	1	1	5.6			18			CFR	Sri Lanka	[8]
carbofuran	3	2	2.9	1.0	4.1	209	100	479	CFR	Brazil, Sri Lanka	[8,9,54]
carbosulfan	2	1	17.1	10.7	23.5	198	51	345	CFR	Sri Lanka	[8,9]
chlorfluazuron	1	1	2.2			45			CFR	Sri Lanka	[8]
chlorpyrifos	7	3	6.2	5.2	8.0	208	34	1376	CFR, GCS, PSS	Brazil, India, Sri Lanka	[7,8,9,23,26,29,54]
cypermethrin	2	2	6.1	5.1	7.0	50	41	58	CFR	Brazil, India	[7,54]
deltamethrin	1	1	0.0			11			CFR	Sri Lanka	[8]
diazinon	1	1	4.8			84			CFR	Sri Lanka	[8]
dichlorvos	2	1	32.3	31.3	33.3	13	9	16	CFR	Japan	[50,75]
dimethoate	6	2	23.6	5.5	30.8	268	17	833	CFR, GCS, PSS	Brazil, Sri Lanka	[8,9,23,26,29,54]
diquat	1	1	0.0			312			CFR	USA	[48]
edifenphos	1	1	11.8			17			CFR	Sri Lanka	[8]
endosulfan	6	3	22.9	20.2	29.3	86	9	400	CFR	Brazil, India, Sri Lanka	[7,8,9,36,49,54]
endrin	1	1	5.0			74			CFR	India	[7]
esfenvalerate	1	1	8.3			12			CFR	Sri Lanka	[8]
etofenprox	1	1	0.8			121			CFR	Sri Lanka	[8]
fenitrothion	2	1	15.4	9.4	21.3	40	32	47	CFR	Japan	[50,75]
fenobucarb	2	1	5.6	5.3	5.8	71	38	104	CFR	Sri Lanka	[8,9]
fenoxaprop-p-ethyl	1	1	0.0			74			CFR	Sri Lanka	[8]
fenthion	4	1	13.9	4.3	16.2	111	23	237	CFR, GCS, PSS	Sri Lanka	[8,9,23,26]
fipronil	1	1	0.0			26			CFR	Sri Lanka	[8]
glufosinate	1	1	7.1			14			CFR	Japan	[50]
glyphosate	16	6	6.1	0.1	29.3	102	15	3464	CFR, PSS	Brazil, Japan, Korea, Sri Lanka, Taiwan, USA	[8,9,18,20,21,38,39,48,50,54,55,59,63,67,68,76]
hydrogen phosphide	1	1	2.6			152			CFR, PSS	Germany	[37]
imidacloprid	2	2	0.0	0.0	0.0	39	8	70	CFR	India, Sri Lanka	[7,8]
indoxacarb	1	1	14.0			7			CFR	India	[7]
lindane	1	1	0.0			3			CFR	Sri Lanka	[8]
malathion	7	5	6.5	0.0	25.0	23	5	209	CFR, APACHE	Brazil, India, Japan, Singapore, Sri Lanka	[7,8,9,40,50,54,75]
MCPA	2	1	5.1	4.8	5.4	387	93	681	CFR	Sri Lanka	[8,9]
metam sodium	1	1	0.0			102			CFR	France	[24]
methamidophos	3	2	12.5	11.5	15.4	26	8	191	CFR	Brazil, Sri Lanka	[8,9,54]
methomyl	2	1	7.2	0.0	14.3	31	7	54	CFR	Sri Lanka	[8,9]
monocrotophos	3	3	22.2	20.4	35.0	99	54	257	CFR	Brazil, India, Sri Lanka	[7,9,54]
oxydemeton-methyl	2	2	13.4	12.5	14.3	11	8	14	CFR, PSS, APACHE, SOFA, GCSCFR	Germany, Sri Lanka	[8,33]
oxyfluorfen	1	1	0.0			15			CFR	Sri Lanka	[8]
paraquat	14	6	54.2	1.4	83.6	115	7	1046	CFR	Brazil, Japan, Korea, Sri Lanka, Taiwan, USA	[8,9,18,20,28,34,35,41,47,48,50,54,73,76]
parathion ethyl	1	1	42.9			7			CFR, PSS, APACHE, SOFA, GCS	Germany	[33]
parathion methyl	1	1	60.0			5			CFR	India	[7]
permethrin	1	1	0.0			13			CFR	Sri Lanka	[8]
petrilachlor	1	1	0.0			11			CFR	Sri Lanka	[8]
phenthoate	2	1	7.4	6.5	8.3	96	24	168	CFR	Sri Lanka	[8,9]
phorate	1	1	19.0			21			CFR	India	[7]
picloram	1	1	25.0			5			CFR	Brazil	[54]
pirimiphos-methyl	1	1	0.0			12			CFR	Sri Lanka	[8]
profenofos	2	1	5.5	0.0	11.0	84	22	146	CFR	Sri Lanka	[8,9]
propamocarb	1	1	100.0			1			CFR	Sri Lanka	[8]
propanil	3	1	5.0	1.6	10.9	150	64	412	CFR	Sri Lanka	[8,9,27]
propoxur	1	1	0.0			16			CFR	Sri Lanka	[8]
prothiofos	1	1	7.7			13			CFR	Sri Lanka	[8]
quinalphos	2	2	12.1	12.0	12.1	101	78	124	CFR	India, Sri Lanka	[7,8]
rotenone	1	1	0.0			54			CFR	USA	[48]
spinosad	1	1	0.0			4			CFR	India	[7]
triazophos	1	1	17.0			6			CFR	India	[7]
trichlorfon	2	1	18.8	0.0	37.5	8	7	8	CFR	Japan	[50,75]
trifuralin	1	1	0.0			17			CFR	Brazil	[54]
zink phosphide	2	1	7.6	4.2	11.0	30	24	35	CFR	India	[49,77]

^1^ CFR: case-fatality-rate, PSS: Poisoning severity score, GSC: Glasgow Coma Score, APACHE: Acute Physiology and Chronic Health Evaluation Score, SAPS: Simplified Acute Physiology Score, SOFA: Sequential Organ Failure Assessment. ^2^ OP: organophosphate.

**Table 2 ijerph-18-08307-t002:** Countries addressed and number of papers providing case-fatality of pesticide poisoning.

Country	No. of Paper	No. of Papers Providing Case-Fatality on	
		Group Level	Active Ingredient Level
Australia	1	1	-
Brazil	3	2	1
China	1	1	-
France	2	-	2
Germany	2	1	2
India	8	7	4
Iran	3	1	2
Israel	4	4	-
Japan	3	1	3
Jordan	1	1	-
Korea	5	1	4
Serbia	1	1	-
Singapore	1	-	1
Slovenia	1	1	-
South Africa	2	2	-
Sri Lanka	10	5	9
Taiwan	12	3	10
Turkey	4	4	-
USA	2	1	2
Zimbabwe	1	1	-
All	67	38	40

e.g., India: 8 papers total with 4 exclusively on group level, 1 exclusively on active ingredient, 3 both.

**Table 3 ijerph-18-08307-t003:** Case fatality ratios for active pesticide ingredients by WHO classification.

WHO Class *	Median Case-Fatality-Ratio									
	<1		1 – <10		10 – <20		≥20%		All	
	*N*	%	*N*	%	*N*	%	*N*	%	*N*	%
**Ia**			1	4	1	8	2	18	4	6
**Ib**			3	13	5	38	2	18	10	15
**II**	11	61	14	58	6	46	4	36	35	53
**III**	3	17	3	13					6	9
**O**			1	4	1	8			2	3
**U**	4	22	1	4			2	18	7	11
**VF**			1	4			1	9	2	3
**All**	18	100	24	100	13	100	11	100	66	100

* Ia = “extremely hazardous”, Ib = “highly hazardous”, II = “moderately hazardous”, III = “slightly hazardous”, U = “unlikely to present acute hazard”, O = “obsolete”, VF = “volatile fumigant not classified” see WHO [6] for details.

**Table 4 ijerph-18-08307-t004:** Variability of case-fatality-ratios (%) for pesticides *.

Name	WHO Class **	Publications	Cases	Case Fatality Ratio					
		*n*	Median	Min	Mean	Median	Max	CV	CV Norm
carbamates		9	60	0	5	5	14	98	35
organophosphate		31	94	3	15	11	73	92	17
chlorpyrifos	II	7	208	5	7	6	8	19	8
dimethoate	II	6	268	6	22	24	31	40	18
endosulfan	II	6	86	20	24	23	29	16	7
fenthion	II	4	111	4	12	14	16	44	26
glyphosate	III	16	102	0	7	6	29	100	26
malathion	III	7	23	0	10	7	25	93	38
paraquat	II	14	115	1	49	54	84	56	16

* only pesticides addressed in more than 3 papers, ** II = “moderately hazardous”, III = “slightly hazardous”, See WHO [6] for details, min = minimum, max = maximum, CV = coefficient of variation, CV norm = normalized CV.

**Table 5 ijerph-18-08307-t005:** Reported indicators for the severity of poisonings.

Indicator	Pesticide Groups	Active Ingredients
APACHE—Acute Physiology and Chronic Health Evaluation Score	8	4
GCS—Glasgow Coma Score	10	11
PSS—Poisoning Severity Score	11	8
SAPS—Simplified Acute Physiology Score	4	1

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
