# Peer review of "Case Fatality as an Indicator for the Human Toxicity of Pesticides—A Systematic Scoping Review on the Availability and Variability of Severity Indicators of Pesticide Poisoning"

_ijerph, 2021, doi:10.3390/ijerph18168307_

Round 1
Reviewer 1 Report
The study is of partial merit for publication. The topic is of high-interest since pesticide poisonings and fatality are still a key problem of modern society. I have several comments that need to be addressed prior consideration of acceptance.
- It is rather risky to incorporate the poisonings and fatality incidents in an overall concept of "accusing" the pesticide formulation, since the majority of cases are suicide attempts. Hence, i suggest the authors to include in their discussion or at least mention parameters and mitigation measures such as training or proper handling, and disposal of containers, as a key step prior "blaming" only the chemical (regardless if it is pesticide etc.). Otherwise, misleading conclusions can be derived (page 2, end of 5th paragraph).
- Page 4, 3rd paragraph please delete the line "the search strategy was developed by both authors".
- In the introduction authors can mention how analytical chemistry helps in the elucidation of poisoning incidents (e.g., add reference, Kasiotis et al., 2021, Open Chemistry, Investigating a human pesticide intoxication incident: The importance of robust analytical approaches, https://doi.org/10.1515/chem-2021-0193)
- A potential reference to extension of the study after 2014 should be mentioned. Maybe some important active substances (authorised after 2013) are missing from the list, and from my perspective is a limitation of the study, especially for European Union and USA.
- In Table 1, please add the year as well.
Author Response
Q1 :It is rather risky to incorporate the poisonings and fatality incidents in an overall concept of "accusing" the pesticide formulation, since the majority of cases are suicide attempts. Hence, i suggest the authors to include in their discussion or at least mention parameters and mitigation measures such as training or proper handling, and disposal of containers, as a key step prior "blaming" only the chemical (regardless if it is pesticide etc.). Otherwise, misleading conclusions can be derived (page 2, end of 5th paragraph).
Q3: In the introduction authors can mention how analytical chemistry helps in the elucidation of poisoning incidents (e.g., add reference, Kasiotis et al., 2021, Open Chemistry, Investigating a human pesticide intoxication incident: The importance of robust analytical approaches, https://doi.org/10.1515/chem-2021-0193)
Ad 1 and 3: We thank the reviewer to point to the role of mitigating factors as well as analytical approaches. We added the recommended reference and the following sentence to the discussion chapter. “Identification of factors influencing the CFR can further be supported by analytical approaches [87]. This might assist to objectively link exposure data to symptoms and allows mitigating factors like protective clothing and handling procedures as well as quality and stability of pesticide formulations to be taken into account.”
Q2: Page 4, 3rd paragraph please delete the line "the search strategy was developed by both authors".
Ad 2: We fail to see why the sentence “The search strategy was developed by both authors” should be deleted. Since the PRISMA-Statement for systematic reviews encourages to detail the roles of authors in all steps of the review we prefer to keep this information in the manuscript. However, if we miss an important argument, we can delete the sentence without any problem.
Q4. A potential reference to extension of the study after 2014 should be mentioned. Maybe some important active substances (authorised after 2013) are missing from the list, and from my perspective is a limitation of the study, especially for European Union and USA.
Ad 4: We agree that possibly missing data for newly registered pesticides could be a limitation of our study. Although we discussed results of a recent publication from 2021, we anticipate a large time lag before poisonings by newly approved pesticides are published. To highlight this, we added the following sentence in the discussion chapter: ”Our review certainly omitted newly authorized pesticides because human case-fatality studies usually are carried out after longer-term use.”
Q5. In Table 1, please add the year as well.
Ad 5: We prefer to refrain from adding the year in table 1 because several numbers would have to be added per line leading to another rather loaded column with redundant information. Furthermore, the publication years can easily be seen from the references which are given in the table.
Reviewer 2 Report
The authors reviewed the scientific literature published between 1990 and 214. Only 67 papers could be included in the assessment indicating one of the major constrains of the review. The authors correctly identified some of the limitation of their study. One aspect which should be additionally considered in future reports and assessment is the quality of pesticides caused by poisoning. A good example for its importance is the malathion. In that case most probably its >thousand times more toxic iso-malathion content, which may be formed during extended storage or derived from improper formulation, was the source of fatal poisoning.
The word “registration” in the last sentence should be changed to periodic review. (During the period of initial registration human fatalities are unlikely occur).
Author Response
We thank the reviewer for pointing to the aspect of chemical quality. We referenced this problem by adding following sentence in the discussion section: „This might assist to objectively link exposure data to symptoms and allows mitigating factors like protective clothing and handling procedures as well as quality and stability of pesticide formulations to be taken into account”
We rephrased referring to “registration” in the abstract as well as in the conclusion part of the manuscript, now reading “… in the periodic statutory review of pesticides and their regulation.”
Round 2
Reviewer 1 Report
The authors addressed the comments raised. Hence, i suggest acceptance of the manuscript in its current state.
Author Response
Dear Reviewer,
Thanks for your comments.
Kind regards.